# Improved Sliding Mode-Active Disturbance Rejection Control of Electromagnetic Linear Actuator for Direct-Drive System

**Yingtao Lu, Cao Tan *** 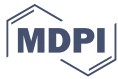**, Wenqing Ge, Bo Li and Jiayu Lu**

School of Transportation and Vehicle Engineering, Shandong University of Technology, 266 West Xincun Road, Zhangdian District, Zibo 255000, China; luyingtaosdut@yeah.net (Y.L.); gwq@sdut.edu.cn (W.G.); libo@sdut.edu.cn (B.L.); lu_j_y@126.com (J.L.)
* Correspondence: njusttancao@yeah.net; Tel.: +86-15264311526

**Abstract:** The electromagnetic linear actuator is used as the core drive unit to achieve high precision and high response in the direct-drive actuation system. In order to improve the response performance and control accuracy of the linear drive unit, an improved sliding mode-active disturbance rejection control (ISM-ADRC) method was proposed. A motor model was established based on improved LuGre dynamic friction. The position loop adopts the improved integral traditional sliding mode control based on an extended state observer, and the current loop adopts PI control. The stability of the system is verified based on the Lyapunov theory. A nonlinear dilated state observer is used to effectively observe the electromagnetic linear actuator position and velocity information while estimating and compensating the internal and external uncertainty perturbations. At the same time, the saturation function sat(s) is used to replace the sign(s) and introduce the power function of the displacement error variable. The improved integral sliding mode control law further improves the response speed and control accuracy of the controller while reducing the jitter inherent in the conventional sliding mode. Simulation and experimental data show that the proposed improved sliding mode-active disturbance rejection control reduces the 8-mm step response time of the electromagnetic linear actuator by 21.9% and the steady-state error by less than 0.01 mm compared with the conventional sliding-mode control, while the system has 49.4% less adjustment time for abrupt load changes and is more robust to different loads and noise.

**Keywords:** electromagnetic linear actuator; direct drive; traditional sliding mode control; active disturbance rejection control; position control





## 1. Introduction

Electromagnetic linear actuators are widely used in lithography machines [1], linear Stirling refrigerators [2], aerospace [3], electro-hydrostatic actuators [4], and other fields. Take the direct-drive hydrostatic actuator as an example. Compared with the traditional hydrostatic actuator driven by a rotating motor, the new high-power density linear motor based on the Halbach permanent magnet array directly drives the piston of the hydraulic pump, which can achieve the advantages of high response, high precision movement, adjustable stroke, low noise, high efficiency, and strong antipollution ability, greatly improving the performance of the electro-hydraulic actuator. The direct-drive control technology will be the main development trend of electrostatic hydraulic actuators in the future and has a wide range of application prospects in the fields of bionic robots, military, intelligent manufacturing, and other fields [5].

The control goal of the electromagnetic linear actuator servo control system is to realize the direct-drive control of nonrepetitive displacement with high efficiency and high precision. The electromagnetic linear actuator directly drives the load but lacks an intermediate buffer link. All kinds of nonlinear and time-varying internal and external interference will directly act on the motor mover without buffering. Especially in complex working conditions, the dynamic and steady-state performance of the controlled system

is greatly affected [6]. Therefore, controllers with good robustness and anti-interference ability have been continuously studied and applied.

In order to improve the tracking accuracy and high-speed action of electromagnetic linear actuators, a variety of control methods have been proposed. Traditional PID control is the most widely used in industrial control because of its simple structure and it does not rely on precise mathematical models. However, it is difficult to meet the high-frequency and high-precision control requirements of multitarget displacement and complex uncertain conditions, and the robustness to parameter time-varying systems is poor [7]. Therefore, modern control methods with robustness have become a research hotspot in the field of electromagnetic linear actuator control, such as sliding mode variable structure control, robust control, ADRC and adaptive control, etc., which have been continuously researched and applied [8].

Traditional sliding mode control, as a nonlinear control method with strong robustness when dealing with system time-varying, nonlinear, and uncertainties, is widely used in motor servo control systems [9], power control systems [10], and electro hydrostatic actuators [11]. However, traditional sliding mode control compensates for uncertainty by using the high gain in the discontinuous control law. High gain inevitably introduces high-frequency chattering to affect the steady-state accuracy and power consumption of the system [12]. In addition, this type of algorithm still has room for improvement in the case of large parameter change ranges and large interference effects. Fan et al. [13] combined the advantages of adaptive control and sliding mode control to achieve precise control of valve motion. The system has good robustness and solves the problems of high response and "soft seating" of valve train. Shi [14] proposed a linear motor point motion control algorithm based on fractional order active disturbance rejection to solve the problem that the linear motor point motion control system is susceptible to interference. Shen proposed the combination of an ADRC and fuzzy controller to solve the problem of adaptive adjustment of controller parameters under different conditions [15]. While improving the anti-interference ability, the fuzzy controller is used to adjust the ADRC parameters online so that the controller can adapt to different working conditions. However, fuzzy control needs to establish a fuzzy rule control table on the basis of experience, and it mostly relies on the experience and knowledge of researchers. In recent years, intelligent algorithms have also been continuously applied to the displacement servo control of electromagnetic linear actuators, but this kind of control system is more complex, requires higher control hardware, and relies on the experience and knowledge of experts [16–18].

In addition, under actual complex working conditions, various uncertain disturbances such as changes in load seriously affect the response quality of the system [19]. Cao [20] designed a disturbance estimator for load resistance and thrust fluctuation and compensated for the disturbance estimated value, which solved the problem of load resistance and thrust fluctuation interference of the control system and significantly improved the positioning accuracy of the motor position. Wang [21] designed an inner loop observation compensation controller based on discrete sliding mode variable structure control, which effectively compensated for the influence of external uncertain disturbances and other factors on system performance. Compared with traditional PID control, it reduces response time and steady-state error while ensuring the robustness of the system. In paper [22], the disturbance observer is combined with a wavelet transform to solve the problem of adaptive control parameters of the observer, compensate for the disturbance of measurement noise, and improve the response quality of the linear servo motor. The extended state observer (ESO) proposed by researcher Han classifies the internal and external factors affecting the system as total interference for real-time estimation and compensation [23]. Due to the fact that the specific model does not depend on the generated disturbance, it has been successfully applied to high-precision industrial control.

In this paper, a double closed-loop composite controller is proposed to satisfy the requirements of high steady-state accuracy, high response speed, and strong robustness of the electromagnetic direct-drive hydrostatic actuator used in the direct-drive electromagnetic

linear actuator servo system. In order to solve the problem of chattering, which leads to system failure and the poor response quality of the controller under complex disturbance conditions, the proposed design combines the advantages of sliding mode control and active disturbance rejection control and solves the contradiction between system rapidity and overshoot through TD. The improved integral sliding mode control law has the advantages of fast response, high precision, and strong anti-interference ability. It solves the chattering problem of traditional sliding mode and improves the control accuracy of the system. At the same time, the ESO can compensate the displacement, internal and external disturbances of the system, solve the limitation of the observation compensation range of the general disturbance observer, filter the noise in the data acquisition, and improve the anti-interference ability of the controller. The simulation and experimental results show that the controller improves the position control effect of the nonlinear system of the electromagnetic linear actuator.

## 2. Working Principle and Modeling of Electromagnetic Linear Actuator

### 2.1. Working Principle of Electromagnetic Linear Actuator

The electromagnetic linear actuator described in this paper is mainly composed of a moving coil assembly, composed of a coil winding and a coil frame, and a Halbach permanent magnet array assembly fixed on a ferromagnetic yoke. The two components are shown in Figure 1. By using a Halbach array with optimized design parameters, the air gap magnetic field is enhanced and the performance of the electromagnetic linear actuator is improved. When the coil is in the air gap magnetic field formed by a high-performance permanent magnet, it is judged by the left-hand rule that it is subjected to by the upward electromagnetic force, and then produces the upward displacement movement. Therefore, the accurate action of the moving coil can be controlled by controlling the coil current and the direction.

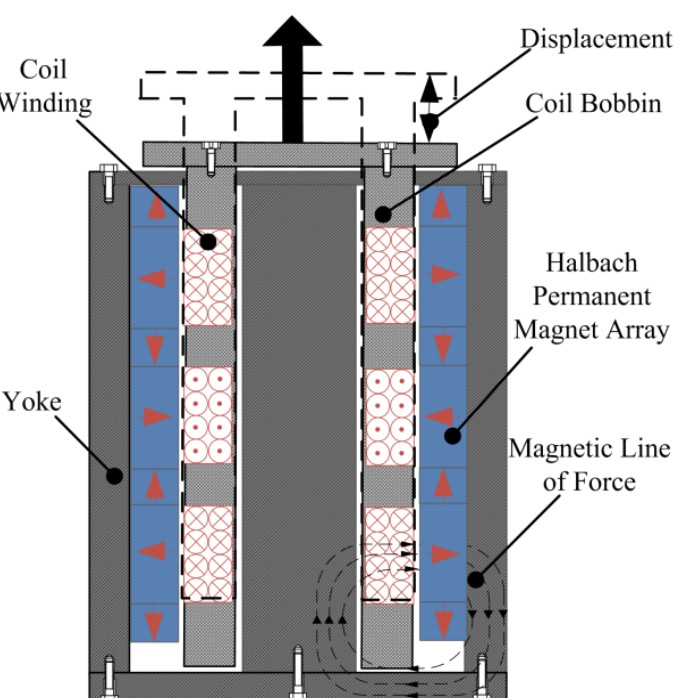

**Figure 1.** Schematic diagram of electromagnetic linear actuator structure.

### 2.2. Kinetic Modeling

An electromagnetic linear actuator is a system of mechanical, electrical, and magnetic subsystems coupling with each other [24] (Figure 2). The mathematical expressions of the electrical subsystem, magnetic circuit subsystem, and mechanical subsystem are as follows:

$$\begin{cases} U(t) = E + RI(t) + L\frac{dI(t)}{dt} \\ E = K_e v \end{cases} \tag{1}$$

$$F_e = NBLI(t) = K_m I(t) \tag{2}$$

$$M\ddot{x} = F_e - F_f - c\dot{x} - F_d \tag{3}$$

where $U$ is the phase voltage; $E$ is the back electromotive force; $R$ is the resistance of the coil; $L$ is the inductance; $I$ is the current; $v$ is the speed of the mover; $K_e$ is the back electromotive force coefficient; $F_e$ is the electromagnetic force; $N$ is the total number of turns of the coil; $B$ is the magnetic field strength; $K_m$ is the electromagnetic force coefficient; $M$ is the mass of the mover of the motor; $x$ is the displacement of the mover; $F_f$ is the friction force received during the movement of the mover; $c$ is the damping coefficient; $F_d$ is internal and external uncertain interference forces.

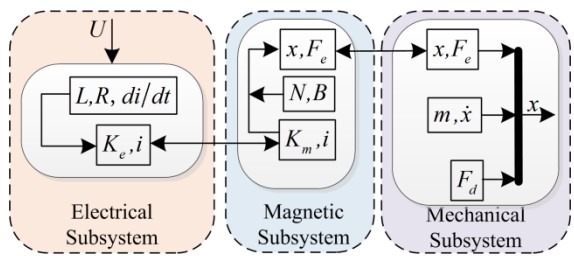

**Figure 2.** Schematic diagram of electromagnetic linear actuator.

The effect of friction on high-performance positioning and low-speed tracking control is particularly obvious. In order to accurately establish the electromagnetic linear actuator model, an improved LuGre dynamic friction model is adopted [25], the expression is as follows:

$$\begin{cases} F_f = \sigma_0 s(v)z + \sigma_1 h(v)\dot{z} + A_f sign(v) \cdot [1 - s(|v|)] + \alpha_2 v \\ \dot{z} = s(|v|)\left(v - \frac{|v|}{g(v)}z\right) \\ g(v) = \alpha_0 + \alpha_1 e^{-(v/v_s)^2} \\ s(|v|) = \begin{cases} 1, |v| < v_1 \\ \frac{1}{v_1 - v_2}(|v| - v_2), v_1 < |v| < v_2 \\ 0, |v| > v_2 \end{cases} \end{cases} \tag{4}$$

where $\sigma_0$ is the stiffness; $\sigma_1 h(v)$ is the "bristle" damping coefficient; $z$ is the immeasurable internal friction state; $\alpha_2$ is the viscous friction coefficient; $v$ is the velocity of the mover; $A_f$ is the Coulomb friction coefficient; $v_s$ is the *Stibeck* velocity; the positive definite function $g(v)$ is used to describe the *Stibeck* phenomenon.

Based on the above mathematical model, the state space equation of the electromagnetic linear actuator can be obtained as:

$$\begin{cases} \frac{dI(t)}{dt} = -\frac{R}{L}i - \frac{K_e}{L}v + \frac{U(t)}{L} \\ \dot{v} = \frac{-F_f - cv}{M} + \frac{K_m}{M}I(t) - \frac{F_d}{M} \\ \dot{x} = v \end{cases} \tag{5}$$

## 3. Control System Design

### 3.1. The Overall Design of the Control System

The controlled system can be transformed into a first-order system and a second-order system in series. A double closed-loop control system of the current loop and the position loop is adopted, and the inner loop current PI is controlled to make the current loop stable without static error and dynamic without overshoot. The outer ring is composed of two parts, a compound position control system based on ISM-ADRC. Compound control combines the respective advantages of SMC and ADRC. The improved integral sliding mode control law is adopted to achieve fast and high-precision response while being robust to disturbances. The ESO observes and compensates for displacement, internal and external disturbances, etc., so as to improve the robustness and anti-interference ability of the controller. This will greatly improve the dynamic and steady-state response performance of the system. The entire control system framework is shown in Figure 3.

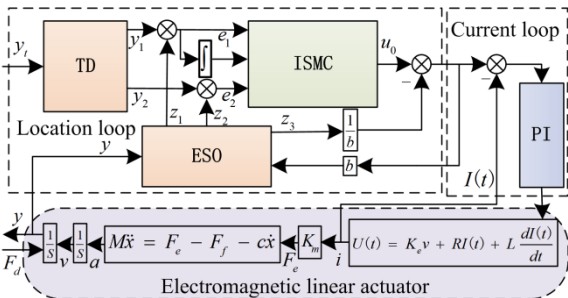

**Figure 3.** Diagram of the control system.

### 3.2. The Design of the Position Loop Controller

Simplifying the electromagnetic linear actuator model to a second-order system, the state equation of its mathematical model is:

$$
\begin{cases}
\dot{x}_1 = x_2 \\
\dot{x}_2 = \frac{K_m}{M}u - \frac{F_f + cv + F_d}{M} \\
y = x_1
\end{cases}
\tag{6}
$$

where $x_1$ is the displacement of the mover; $x_2$ is the speed of the mover; $u$ is the coil current.

#### 3.2.1. Design of Improved Integral Sliding Mode Control Law

Sliding mode control is a discontinuous variable structure control with strong robustness, but discontinuous switching control is accompanied by high-frequency chattering [25,26]. In order to reduce the steady-state error caused by high-frequency chattering, this design adopted the improved sliding mode control (ISMC) to achieve fast, chatter-free, and accurate response. The position error of the electromagnetic linear actuator servo system is defined as follows:

$$
e_y = y_t - y
\tag{7}
$$

where $y_t$ is the target input of the control system; $y$ is the displacement of the controlled actuator.

The steady-state error is further reduced by introducing the integral term of the tracking error. The designed integral sliding mode surface is:

$$
s = k_1 e_y + \dot{e}_y + k_2 \int_0^\tau e_y dt
\tag{8}
$$

where $k_1$ and $k_2$ are controller parameters.

By deriving the above Formula (8), we can get:

$$
\begin{aligned}
\dot{s} &= k_1\dot{e}_y + \ddot{e}_y + k_2e_y \\
&= k_1\dot{e}_y + \ddot{y}_t - \frac{K_m}{M}u + \frac{F_f(\dot{y})+c\dot{y}}{M} + k_2e_y
\end{aligned} \tag{9}
$$

In order to ensure the rapid stability of the sliding mode motion process and reduce the system chattering, the reaching law design of this controller adopts the saturation function $sat(s)$ instead of the $sign(s)$, and at the same time introduces the power of the $|e_y|$ [27]. The improved exponential reaching law is then designed as follows:

$$
\begin{cases}
\dot{s} = -\xi|e_y|^a sat(s) - \eta s \\
\lim\limits_{t\to\infty}|e| = 0, a \geq 0, \eta > \xi > 0
\end{cases} \tag{10}
$$

where $sat(s)$ is the saturation function, which is defined as follows:

$$
sat(s) = \begin{cases}
1 & s > \Delta \\
s & \Delta|s| \leqslant \Delta \\
-1 & s < -\Delta
\end{cases} \tag{11}
$$

where $\Delta$ is the thickness of the boundary layer.

The control variable $u$ obtained by combining (9) and (10) and simplifying is:

$$
u = \left[k_1\dot{e}_y + \ddot{y}_t + \frac{F_f(\dot{y})+c\dot{y}}{M} + k_2f(e_y) + \xi|e_y|^a sat(s) + \eta s\right] \cdot \frac{M}{K_m} \tag{12}
$$

3.2.2. Design of the Active Disturbance Rejection Controller

This controller comprehensively considers factors such as the maximum acceleration that the electromagnetic linear actuator can withstand and external loads. It arranges the appropriate transition process by using the tracking differentiator (TD). The differential tracker can be expressed as [28,29]:

$$
\begin{cases}
\dot{y}_1 = y_2 \\
\dot{y}_2 = fhan(y_1(k) - y_t(k), y_2(k), r, h_0)
\end{cases} \tag{13}
$$

where $r$ and $h_0$ are the parameters of the tracking differentiator, and express $fhan(y_1(k) - y_t(k), y_2(k), r, h_0)$ as:

$$
\begin{cases}
d = rh_0{}^2 \\
a_0 = h_0y_2(k) \\
y_0 = y_1(k) - y_t(k) + a_0 \\
\quad a_1 = \sqrt{d^2 + 8d \cdot |y_0|} \\
a_2 = a_0 + \frac{sign(y_0)\cdot(a_1-d)}{2} \\
a = (a_0 + y)fsg(y_0,d) + a_2(1 - fsg(y_0,d)) \\
fhan = -r(\frac{a}{d})fsg(a,d) - r \cdot sign(a)(1 - fsg(a,d)) \\
fsg(y_0,d) = (sign(y_0 + d) - sign(y_0 - d))/2
\end{cases} \tag{14}
$$

Considering that the system is affected by uncertainties and external disturbances, the ESO is used to estimate the position loop and the estimated "unknown disturbance" is compensated for by improving the integrated traditional sliding mode control law, according to the second-order system of Equation (6):

$$
f(x, v) = -\frac{F_f + cv}{M} - \frac{F_d}{M} \tag{15}
$$

where $f(x, v)$ is the sum of the disturbances inside and outside the system. Extend the total disturbance into a new state variable $x_3$ and we can make $\dot{x}_3 = \omega(t)$, $u = u_0 - z_3/b$, expand it into an integral series system:

$$
\begin{cases}
\dot{x}_1 = x_2 \\
\dot{x}_2 = bu + x_3 \\
\dot{x}_3 = \omega(t) \\
y = x_1
\end{cases}
\tag{16}
$$

Then the discrete extended state observer can be designed as:

$$
\begin{cases}
e = z_1 - y \\
\dot{z}_1 = z_2(k) - \beta_{01}e \\
\dot{z}_2 = z_3(k) - \beta_{02}fal(e, 0.5, \delta_1) + b_0u \\
\dot{z}_3 = -\beta_{03}fal(e, 0.25, \delta_2)
\end{cases}
\tag{17}
$$

where $z_1$ and $z_2$ are the tracking signal of position feedback and its derivative, and the extended state quantity $z_2$ can estimate the unknown disturbance of the control system; $\beta_{01}, \beta_{02}, \beta_{03}$ is the gain parameter of the observer. $\delta_1$ and $\delta_2$ are the adjustable parameter of the fal function in the extended state observer. The power function expression is defined as:

$$
fal(e, \alpha, \delta) = \begin{cases}
|e|^{\alpha}sign(e), |e| > \delta \\
e/\delta^{1-\alpha}, |e| \leq \delta
\end{cases}
\tag{18}
$$

where $\alpha$ satisfies condition $\alpha < 1$; $\delta$ is a positive integer.

### 3.3. Stability Analysis of Controller

We set the reference target; the error definition model between the integral series system (16) and the observations obtained by the extended state observer (17) are as follows:

$$
\begin{cases}
e_1 = z_1 - y \\
\dot{e}_1 = e_2 - \beta_{01}e_1 \\
\dot{e}_2 = e_3 - \beta_{02}fal(e_1, 0.5, \delta) \\
\dot{e}_3 = \omega - \beta_{03}fal(e_1, 0.25, \delta)
\end{cases}
\tag{19}
$$

It can be seen from the above structure that when the disturbance $\omega$ is zero, the zero point is its equilibrium point, and the system error (19) of the third-order ESO is asymptotically stable at the zero equilibrium point. Formula (7) is further expressed as $\dot{e} = -A(e)e$, and the existence of a third-order square matrix $D$ is defined as follows.

$$
D = \begin{bmatrix}
d_{11} & d_{12} & d_{13} \\
d_{21} & d_{22} & d_{23} \\
d_{31} & d_{32} & d_{33}
\end{bmatrix}
\tag{20}
$$

where $d_{11}, d_{22}, d_{33}$ are greater than zero. When $\beta_{01}, \beta_{02}, \beta_{03}$ are greater than zero and $\beta_{01} \cdot \beta_{02} > \beta_{03}$, the matrix $DA(e)$ can be made positive definite symmetric, and the zero solution of $\dot{e} = -(A(e)e)$ is Lyapunov asymptotically stable [30]. We define $d_{13} = -\varepsilon$, then select the Lyapunov function and expand it.

$$
\begin{aligned}
V &= \int_0^t (DA(e)e, \dot{e})d\tau \\
&= \int_0^t (DA(e)e, -A(e)\dot{e})d\tau \\
&= \int_0^t -(A(e)e)\prime D(A(e)e)d\tau \\
&= \int_0^t -(\beta_{01}e_1 - e_2)^2 - \varepsilon(\beta_{02}fal - e_3)^2 - \varepsilon(\beta_{03}fal)^2 d\tau
\end{aligned}
\tag{21}
$$

Equation (21) only contains the constant elements of the diagonal of the square matrix $D$. It has nothing to do with other elements. The derivative of the Lyapunov function is

less than zero, and the Lyapunov function will always decrease. Therefore, the error of the third-order ESO is stable with respect to the zero equilibrium point.

According to the sliding mode surface selected by the improved sliding mode controller, the Lyapunov function can be taken as:

$$V = \frac{s^2}{2} \tag{22}$$

Take the derivative of Equation (13) and bring in Equation (10) to get:

$$\begin{aligned}
\dot{V} &= s \cdot \dot{s} = s \cdot \left( \xi |e_y|^a sat(s) - \eta s \right) \\
&= -\xi |e_y|^a sat(s)s - \eta s^2
\end{aligned} \tag{23}$$

When $\eta > \xi > 0$ and $s \geq 0$, $-\xi |e_y|^a sat(s)s \leq 0$ and $-\eta s^2 \leq 0$, we can launch $\dot{V} < 0$ at this time. When $s < 0$, the same principle can be obtained, that is $\dot{V} < 0$. The improved sliding mode control law is progressively stable. The specific proof that the tracking differentiator is stable in a finite time is given in paper [28]. Due to space limitations, $i$ will not repeat it. It can be seen from the above that the designed controller is stable.

## 4. Experimental Verification

### 4.1. The Experimental Platform and Prototype

In order to verify the effectiveness of the proposed controller for servo control of electromagnetic linear actuators, an electromagnetic linear actuator experimental platform was built, as shown in Figure 4a. In this experimental platform, a 32-bit floating-point digital signal processor TMS320C28346 with a main frequency of 300 MHz is used as the digital controller. PM11 series position sensors with a mechanical stroke of 11.5 mm, resolution of 0.01 mm, and bandwidth of 2 kHz are used to provide position feedback, and a closed-loop Hall current sensor is used for current measurement. As shown in Figure 4b, the controller collects and transmits signals with the current sensor, position sensor, and H-bridge power drive module through the wiring of the peripheral board and processes the data received by the DSP through the ethernet communication with the personal computer. The power driver is a PWM amplifier. The experimental sampling step is set to 20 kHz. The effective stroke of the electromagnetic linear actuator is 10 mm, and the performance parameters are shown in Table 1.

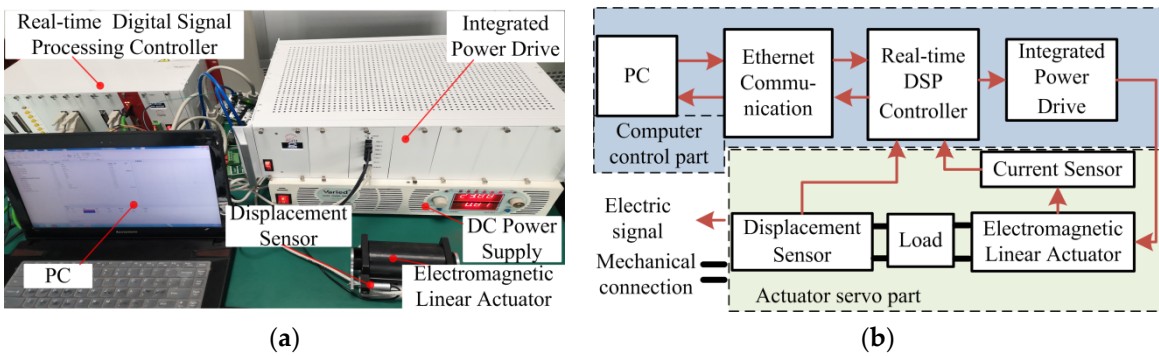

(**a**)    (**b**)

**Figure 4.** Physical map and schematic diagram of the experimental platform: (**a**) experimental platform of electromagnetic linear actuator; (**b**) schematic diagram of electromagnetic linear actuator control system.

**Table 1.** Main performance parameters of electromagnetic linear actuator.

| Parameter | Symbol/Unit | Value |
|---|---|---|
| Coil inductance | L/mH | 1.1 |
| Resistance | R/$\Omega$ | 1.4 |
| Mass of moving part | $M$/kg | 0.12 |
| Back EMF constant | $K_e$/Vs·m$^{-1}$ | 24.61 |
| Force constant | $K_m$/N·A$^{-1}$ | 24.61 |

*4.2. Analysis of Step Target*

The experimental results are shown in Figure 5. The traditional sliding mode controller was used to simulate and compare with the compound controller proposed in this paper. The traditional sliding mode control chooses sliding mode control based on the exponential reaching law. By simplifying the actuator to a second-order system, the control law of Formula (24) can be obtained. As shown in Figure 5a, under the condition of an 8 mm step signal, the simulation results show that two different control algorithms can respond quickly without overshoot tracking. Compared with the traditional sliding mode controller, the response quality of ISM-ADRC has higher steady-state accuracy and faster response speed. ISM-ADRC can better solve the contradiction between the rapidity and stability of displacement tracking. The experimental results show that the traditional sliding mode controller has a certain degree of overshoot, while ISM-ADRC has almost no overshoot and has a fast response. The response time is also close to the simulation result, which verifies the effectiveness of the controller in the simulation test.

$$\begin{cases} \dot{s} = -\xi sgn(s) - ks \\ u = \left[ c\dot{e}_y + \ddot{y}_t + \xi sgn(s) + ks \right] \cdot \frac{M}{K_m} \end{cases} \tag{24}$$

where $\xi, k$ is greater than zero.

The response times of the traditional sliding mode controller simulation and experiment under the step target were 12.6 ms and 15.5 ms, respectively. The positional steady-state error was less than $\pm 0.01$ mm, 0.03 mm, the peak value was 8.6 mm at 12.75 ms, and the overshoot was less than 7.5%. The simulation and experimental response times of ISM-ADRC were 11.05 ms and 12.1 ms, respectively. Compared with the traditional sliding mode controller, times were reduced by 12.3% and 21.9%, respectively. The steady-state error of the simulation position was 0 mm, and the experimental steady-state error was less than 0.01 mm. Compared with the experimental steady-state error of the traditional sliding mode controller, it was reduced by 66.7%. The comparison of experimental results is shown in Table 2. The comparison of simulation results is shown in Table 3. Due to the limitation of the DC power supply voltage, the influence of the fixed connection of the position sensor and the electromagnetic linear actuator, and the limitation of the measurement accuracy of the position sensor, the actual response time and steady-state accuracy values are somewhat different from the simulation results, but they can still meet the response of the drive unit. In summary, the comparison between the above and the curves shows that ISM-ADRC is generally better than SMC in response speed, control accuracy, and stability.

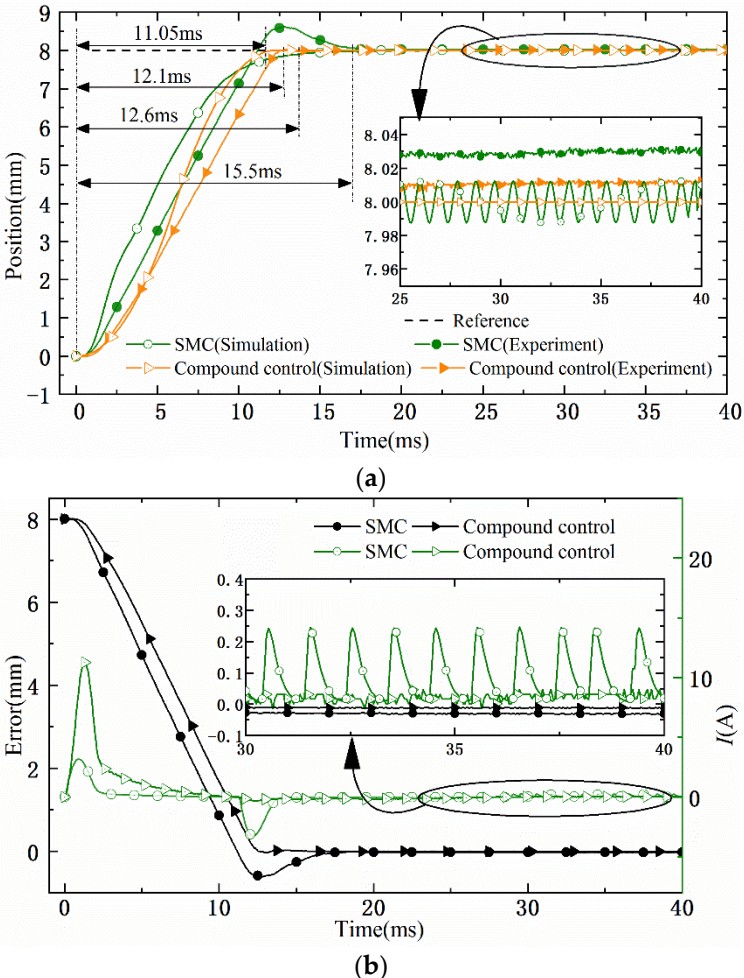

**Figure 5.** Response curve of step under different control: (**a**) response curve of 8 mm step displacement; (**b**) error and current response curve of 8 mm step response displacement.

**Table 2.** Comparison of experimental results under different control conditions.

| Controller | Response Time | Steady-State Error |
|---|---|---|
| SMC | 15.5 ms | 0.03 mm |
| Compound control | 12.1 ms | 0.01 mm |

**Table 3.** Comparison of simulation results under different control conditions.

| Controller | Response Time | Steady-State Error |
|---|---|---|
| SMC | 12.6 ms | $\pm 0.01$ mm |
| Compound control | 11.05 ms | 0 mm |

Figure 5b shows the experimental displacement following error and current response curves of these two controllers. When the electromagnetic linear actuator starts to accelerate, the actuator needs to overcome the influence of non-linear unordered factors such as friction and damping. At the same time, because the current in the energized coil cannot change suddenly, the initial error changes little; but as the current increases, the coil mover's ampere force acts to accelerate the movement, and the position error begins to decrease rapidly. However, due to the influence of the back electromotive force in the coil, the current begins to decrease after reaching the maximum value and finally approaches a steady state, and the following error also approaches zero. However, there is overshoot in

the displacement of the electromagnetic linear actuator under the traditional sliding mode controller. When the current direction increases in the reverse direction and then decreases, it approaches the final steady state, but the current still has a certain frequency of chattering. The error curve and the experimental response curve of Figure 5a can also be seen to have small chattering. This is due to the discontinuous switching of the sign function sign(s) in the traditional sliding mode controller. This not only reduces the steady-state accuracy and response quality of the electromagnetic linear actuator, but also increases the energy loss during actuation.

As shown in Figure 6, in order to test the response performance of the control system to the continuous nonrepetitive step target displacement, 3 mm and 5 mm continuous step targets were designed. It can be seen from Figure 6a that both control algorithms can follow the target displacement better. However, in the experiment, SMC had a certain overshoot. At the same time, there was a big gap in the response speed and steady-state accuracy of the simulation and experiment compared with ISM-ADRC. The simulation and experimental response times of the first step target of ISM-ADRC were 7.7 ms and 8.8 ms, respectively, and the position error was 0 mm, 0.02 mm. The simulation and experimental response times of the second step target were 29.3 ms and 30.1 ms, respectively, and the position error was 0 mm, 0.04 mm. The response times of the traditional sliding mode controller in the simulation and experiment of the first step target were 8.9 ms and 11.4 ms, respectively, the position error was ±0.01 mm, 0.09 mm, and the overshoot was less than 9%. In the second step target, the response times of simulation and experiment were 30.75 ms and 32.95 ms, respectively, the position error was ±0.01 mm, 0.09 mm, and the overshoot was less than 8%. In summary, the simulation and experimental results show that the performance of ISM-ADRC in terms of steady-state accuracy and response speed is greatly improved compared to the traditional sliding mode controller. At the same time, the response results of the ISM-ADRC simulation and experiment are in good agreement, which verifies the accuracy of the simulation model.

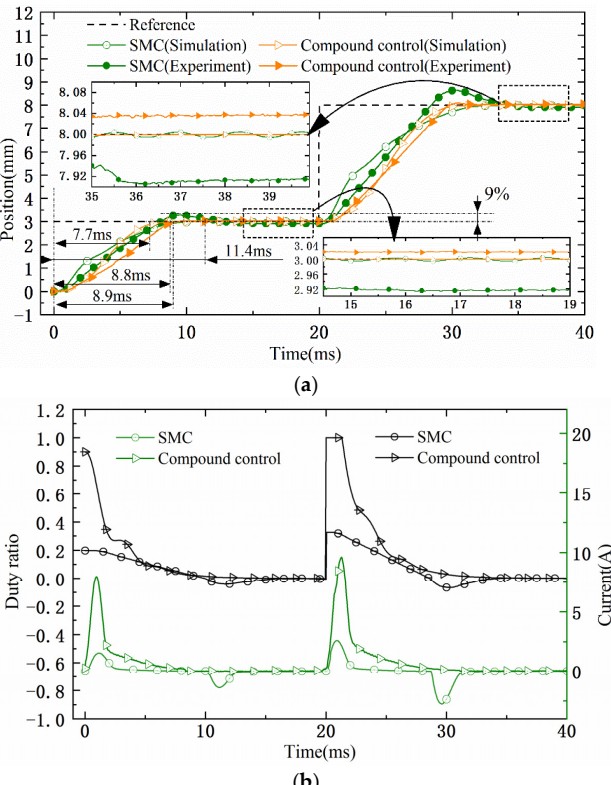

**Figure 6.** Displacement curve of continuous non-repetitive step response: (**a**) displacement curves of different controllers; (**b**) duty ratio and current curve of different controllers.

Figure 6b shows the voltage duty cycle signals and current response curves of the two controllers. When the initial displacement error of the electromagnetic linear actuator is large, the duty cycle is large at the initial stage under the feedback of the controller, and then decreases with the movement of the actuator. At the same time, the current in the energized coil begins to increase. Due to the influence of back EMF in the coil, the current begins to decrease after reaching the maximum value. However, under the sliding mode control, the displacement of the electromagnetic linear actuator has overshoot. When the current direction increases from the opposite direction and then decreases, it approaches the final steady state. At the same time, due to the different responses of the two controllers to the same target, the duty cycle of the two controllers is different, and the peak value of the current is also different. Due to the different goals of the two steps, the peak current of the second step is slightly larger than that of the first step.

## 5. Results and Analysis

In order to verify the response performance of the proposed control method under actual complex working conditions, three different reference target commands under different working conditions were considered. ISM-ADRC was verified and analyzed from two major aspects, response steady-state accuracy and disturbance immunity.

### 5.1. Analysis of System Steady State Accuracy

The response performance and control steady-state accuracy of the proposed control to different loads under the same step target displacement were tested. Figure 7 shows the response curve of a 9 mm step signal under different load conditions. It can be seen from Figure 7a that under the load conditions of 0 N, 10 N, 20 N, 30 N, the ISM-ADRC algorithm of the electromagnetic linear actuator can quickly make tracking without overshoot. The response times were 12 ms, 12.05 ms, 12.1 ms, 12.1 ms, results in good agreement with the tracking curves of different loads. The steady-state error under the maximum load was kept within 1%. Figure 7b shows the displacement estimation curve of the extended state observer and the error curve with the actual displacement under the step response conditions under different loads. It can be seen that the large change in the displacement estimation error at the initial moment was due to the different load. However, under the effective observation of the extended state observer, the change of the error curve approaches 0, and the maximum displacement observation error is much less than 1 μm. Figure 7c shows the step response under different loads, the speed estimation of the extended state observer and the error curve with the actual speed. The speed estimation curves of different loads are in good agreement, and the maximum estimation error of the speed was less than 0.01 m/s, which is within a reasonable range.

Figure 7d shows the estimated response curves of the internal and external disturbances of the ESO under different load conditions. It can be seen that the estimated curve of comprehensive disturbance first increases and then decreases, and the value is much larger than the fixed load. This is due to large nonlinear forces such as friction in low-speed motion, so the estimated compensation value changes greatly. The final estimated value tends to be stable. Although the steady-state value is slightly larger than the value of each fixed load, it can be seen that it is in good agreement with different loads. The ESO not only observes and compensates the external fixed load of the motor, but also observes and compensates for nonlinear factors such as friction and damping inside the motor and the external nonlinear interference. In summary, it can be seen that the ESO has a good observation effect.

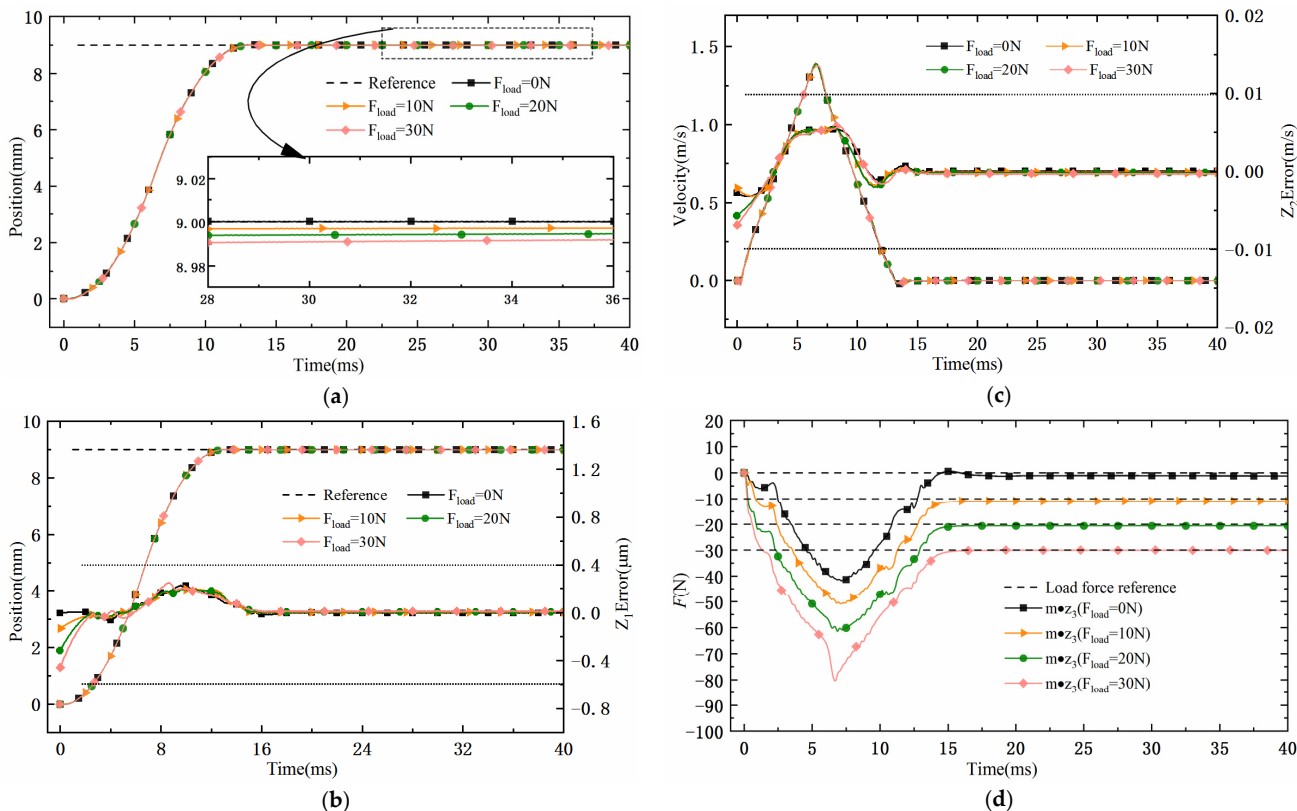

**Figure 7.** The curve of step response under different algorithms: (**a**) response curves of 9 mm step displacement under different load conditions; (**b**) the displacement response and error curve of the ESO under different load conditions; (**c**) speed response and error curve of the ESO under different load conditions; (**d**) response curve of the ESO disturbance estimation under different load conditions.

In the 8 mm, 22 Hz square wave signal condition, the electromagnetic linear actuator responded to the square wave signal, as shown in Figure 8a. Both SMC and ISM-ADRC could achieve better tracking of the square wave signal of the system, and the steady-state error was within a reasonable range. However, ISM-ADRC took a shorter time to enter the steady state in both positive and negative transitions than SMC. As shown in Figure 8b, the steady-state error of the ISM-ADRC target following approached 0, which is much smaller than traditional sliding mode control. The traditional sliding mode control will have small chattering after reaching the positive and negative target values. This is due to the discontinuous switching in the traditional sliding mode controller. Due to the imbalance of the linear drive machinery and the lack of consideration of seating control, the response performance of the two controllers on the rising edge and falling edge was inconsistent. Among them, the response times of ISM-ADRC at the rising and falling edges of the square wave signal were 11.1 ms and 11.4 ms, respectively, and the steady-state error was less than 0.006 mm. SMC required 12.3 ms and 12.5 ms, respectively, and the steady-state error was within 0.017 mm. Compared with SMC, the steady-state error of improved sliding mode-active disturbance rejection control was reduced by 96%. This is because it uses the saturation function sat(s) and improves the reaching law to further weaken the chattering of the system. At the same time, the integration term is introduced to effectively reduce the steady-state error of the system.

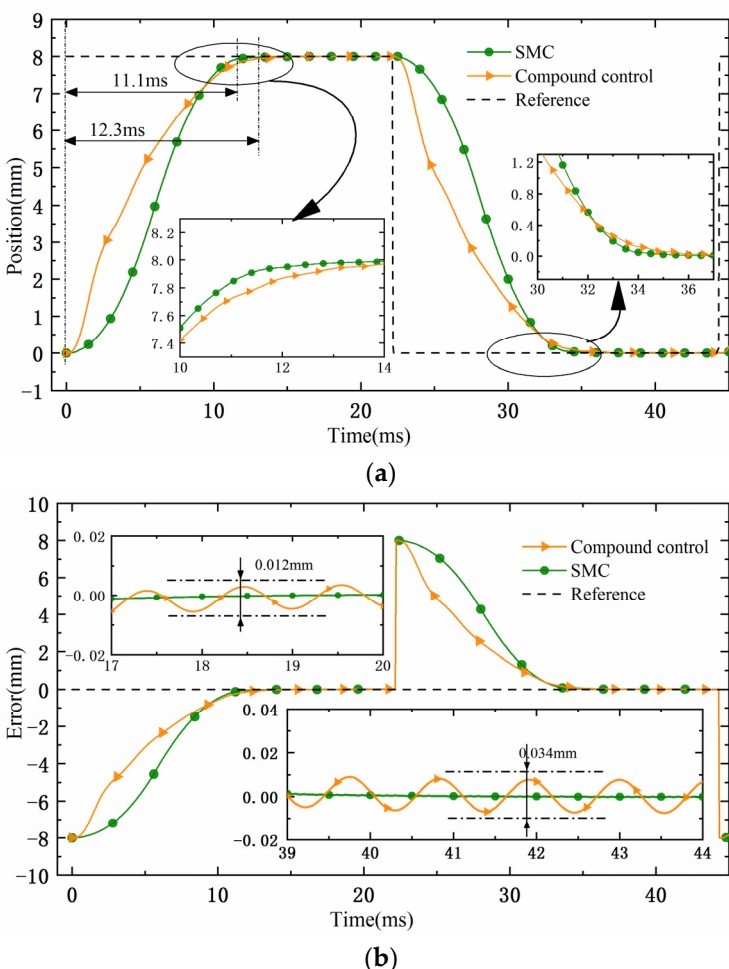

**Figure 8.** Square wave signal response curve under different control algorithms: (**a**) displacement response of wave target under different control algorithms; (**b**) error curve of different control algorithms.

## 5.2. Analysis of System Anti-Disturbance Ability

Taking into account the complex working conditions, the actual various nonlinear and time-varying internal and external interference forces of the electromagnetic linear actuator will directly act on the mover. At the same time, it is also affected by sensor acquisition and command signal noise during operation. In order to verify the anti-interference capability of the control system and the error compensation ability of the ESO under different target displacements, Figure 9 shows the input target position 10 mm step signal, the load changes from 0 N to 40 N at 30 ms, and the system response and position tracking curves under different control algorithms.

Figure 9 shows that the electromagnetic linear actuator stabilized at 10 mm quickly deviated from the target displacement due to the sudden increase of 40 N load disturbance at 30 ms. At the same time, under the action of the extended state observer compensation and feedback control, it quickly returned to the target displacement and maintained stability. In the process of feedback control, the maximum displacement offset of ISM-ADRC is significantly smaller than that of SMC, and it can quickly return to a stable state in a short time. The adjustment time of ISM-ADRC was 2.1 ms, and the overshoot value was less than 0.05%. However, SMC generated a relatively large value of chattering after a sudden load and then slowly decayed to a steady state. The maximum amplitude was 0.04 mm and the adjustment time was 8.3 ms. Compared with ISM-ADRC, the adjustment time was reduced by 49.4%. The results show that ISM-ADRC has good position tracking

performance and still has good robustness and anti-interference ability under large sudden load disturbances.

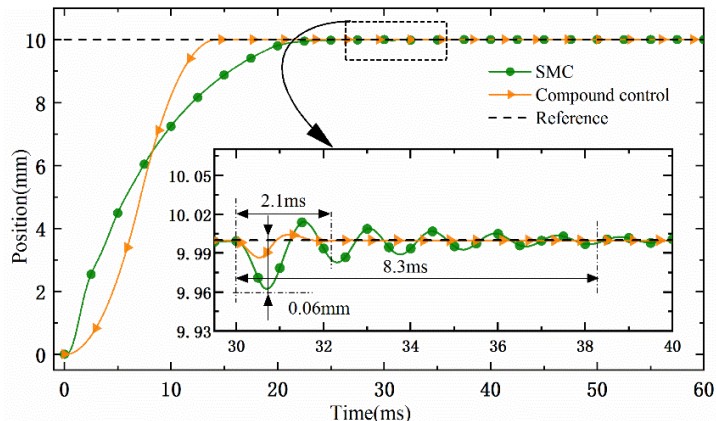

**Figure 9.** Response curves of sudden load under different algorithms.

It can be seen from Figure 10 that the extended state observer has a good function of tracking position, velocity trajectory, and internal and external disturbance estimation. The displacement and velocity observation error of the observer were within a reasonable range. The displacement observation error of the controlled system was less than 0.4 μm and the error of velocity observation was within 0.01 m/s. At the same time, in the case of sudden load disturbance, the ESO could better observe and compensate for the sudden increase of 40 N disturbance, which improved the anti-disturbance ability of the control system against sudden load.

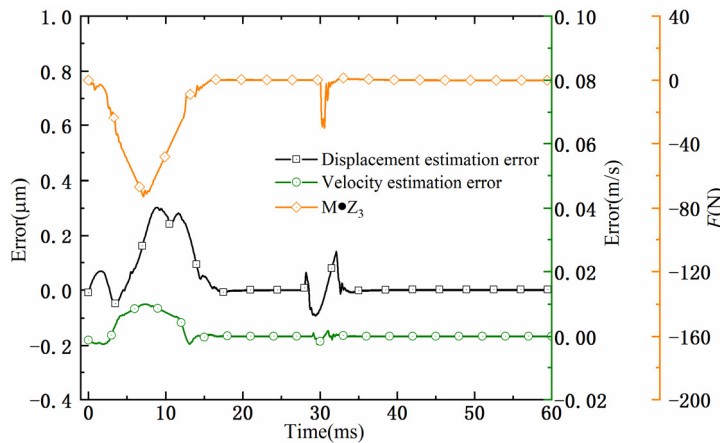

**Figure 10.** Observation error curve of extended state.

In order to test the tracking performance of the system to $y_t = 0.003 sin(\pi t/300) + 0.005$, the input command is a sine wave signal to follow the no-load displacement. The trajectory tracking comparison simulation results of different control algorithms are shown in Figure 11a. At the same time, it is to verify the performance of the servo control system for the noise and external uncertain disturbance of the sensor acquisition signal, and the disturbance compensation performance. Figure 11b shows the comparison curve of sine trajectory tracking and the corresponding error curve under the interference of random white noise with a certain power applied at 20 ms.

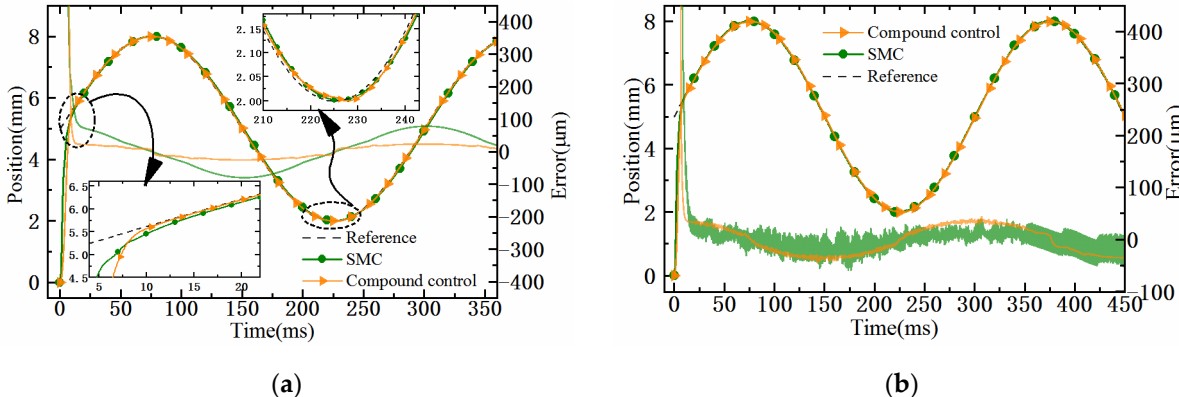

**Figure 11.** Sine trajectory tracking curve under different algorithms: (**a**) displacement response curve of sinusoidal target;
(**b**) displacement response and error of sinusoidal target under white noise.

It can be seen from the position trajectory following in Figure 11a that ISM-ADRC responded to the target displacement faster than SMC. Both control algorithms can follow the response well, but at the same time inevitably have different degrees of lag response. At the second peak, the phase lag time of SMC was 2.65 ms. The phase lag time and relative error of ISM-ADRC were 1 ms and 0.06%, and the lag time was reduced by 62.2% compared with SMC. In the whole tracking process, the tracking error of ISM-ADRC was significantly smaller than the tracking error under SMC, and the steady-state error of ISM-ADRC was much less than 0.02 mm. The comparison shows that ISM-ADRC can not only improve the rapid response performance of the system, but also can track the continuous position command with higher precision and achieve smooth tracking. From Figure 11b, it can be seen that under the condition of adding white noise interference, the anti-interference ability of ISM-ADRC is better than that of SMC. ISM-ADRC has a fast response and good anti-interference and stability, which shows that the ESO has obvious performance in compensating for white noise disturbance and maintains good position tracking performance.

Through the above comparison and analysis, ISM-ADRC can effectively overcome the influence of nonlinear friction, external disturbance and uncertainty, and still has strong robustness and anti-interference performance while responding quickly.

### 5.3. Bandwidth Analysis of Compound Controller Closed Loop System

In a motion control system, it is necessary to discuss the bandwidth of the system. In the design of the control system, we considered the control requirements of the servo system under complex conditions and selected the controller parameters appropriately to maintain the dynamic characteristics and anti-interference performance of the system. From the frequency response diagram of the closed-loop control system, it can be concluded that in the range of 10 Hz, the amplitude response gain was basically zero, and the dynamic performance of the electromagnetic linear actuator was only phase lag. As the frequency increased, the amplitude frequency response and phase frequency response decreased. For the electromagnetic linear servo control system, the closed-loop bandwidth of the control system was 76 Hz, and the phase lag was 35 degrees. The frequency response of the composite controller closed-loop system is shown in Figure 12.

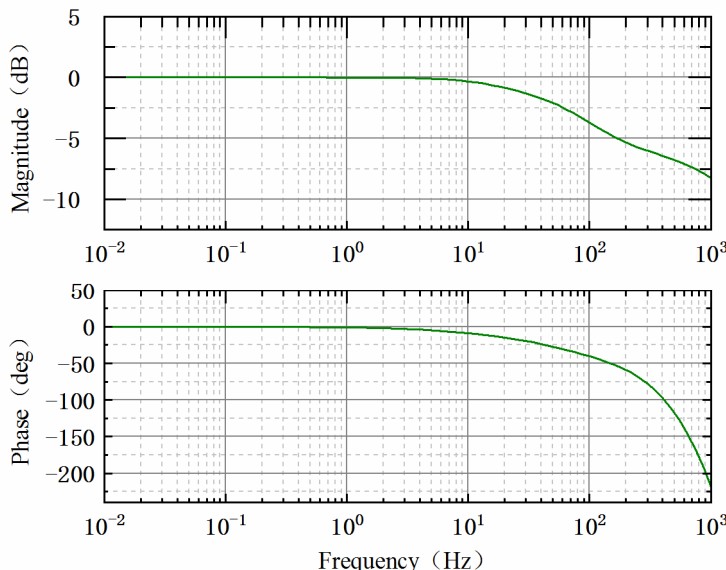

**Figure 12.** Frequency response of closed loop system with compound controller.

## 6. Conclusions

An improved sliding mode-auto disturbance rejection control method is proposed to improve the response performance and control accuracy of electromagnetic linear actuators. It is of great significance for the study of electromagnetic linear actuator servo system controls. After comparing ISM-ADRC with SMC, the following conclusions could be drawn.

(1) ISM-ADRC keeps the original characteristics of high control accuracy and strong anti-interference ability of the active disturbance rejection controller while making full use of the characteristics of fast response and excellent dynamic performance of sliding mode control. Under the input step signal and continuous nonrepetitive step target, it can still quickly make tracking responses without overshoot and chattering. Among them, the step target experiment response time is reduced by 21.9% compared with the traditional sliding mode control, and the steady-state error is less than 0.01 mm, which improves the control system response while ensuring the high precision requirements of the control system.

(2) The nonlinear expansion observer can estimate and compensate the influence of internal disturbance and external disturbance online in real time, and the observation error of displacement and velocity is within a reasonable range. At the same time, the control system can operate under different load conditions, sudden increases of 40 N load disturbance, and system noise. It still has a short dynamic adjustment time and high steady-state accuracy. It also shows good robustness and anti-interference performance, which can meet the higher control accuracy, response speed, and system stability of the electromagnetic direct-drive hydrostatic servo system. The controller has certain application prospects.

**Author Contributions:** Conceptualization, Y.L., C.T. and W.G.; methodology, Y.L., C.T. and B.L.; software, J.L.; validation, Y.L., C.T. and J.L.; formal analysis, Y.L.; investigation, Y.L. and J.L; resources, C.T.; data curation, C.T., W.G. and B.L.; writing—original draft preparation, Y.L.; writing—review and editing, Y.L, C.T. and W.G.; visualization, B.L. and J.L.; supervision, C.T. and J.L.; project administration, C.T., W.G. and B.L. All authors have read and agreed to the published version of the manuscript.

**Funding:** This research was funded in part by the National Natural Science Foundation of China (Grant No. 51905319; Grant No. 51875326; Grant No. 51975341), in part by the China Postdoctoral Science Foundation (Grant No. 2021M691984), and in part by the Major Scientific and Technological Innovation Project of Shandong Province (Grant No. 2019TSLH0703).

**Institutional Review Board Statement:** Not applicable.

**Informed Consent Statement:** Not applicable.

**Data Availability Statement:** Not applicable.

**Conflicts of Interest:** The authors declare no conflict of interest.

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
