# Peer review of "Improved Sliding Mode-Active Disturbance Rejection Control of Electromagnetic Linear Actuator for Direct-Drive System"

_actuators, doi:10.3390/act10070138_

Round 1

Reviewer 1 Report

In the paper the improved SM-ADRC of linear electromagnetic actuator is introduced. In the Introduction section extensive literature review is presented. The controller design is also described in detail and the asymptotic stability of controlled system is proven using Lyapunov function. The theoretical results are proven through simulation and experiment and compared with classical SMC.

However, there are few issues that should be corrected:

  1. Row 84. Could you please explain the meaning of a "load resistance"?
  2. Figure 5(a) and 5(b) introduce important simulation and experimental results of proposed Compound controller, as well as simulation and experimental results of classical SMC. I would recommend to enlarge these Figures.
  3. Result analyses for Figures 5 - 8 would be clearer and more concise if presented in form of tables. Generally, all figures containing simulation and experimental results should be enlarged.
  4. The results in Figures 7(b) and 7(c) are not clearly presented.

Reviewer 2 Report

  1. The motivation behind the article isn't clear. There is nothing in this article that has not been proposed before in the past.
  2. The authors are advised to refer to recent developments in SMC by referring to various recent articles published, e.g, An Improved Sliding Mode Current Control of Induction Machine in Presence of Voltage Constraints, A novel mixed cascade finite-time switching control design for induction motor etc.
  3. The reason for choosing a PID sliding surface isn't clear. And moreover, the authors could have chosen a terminal surface instead of a linear one. 
  4. The signum function is compromised with saturation approximation which will impact its robustness properties. Mooreoberm, the chosen reaching law is a continuous one and we know that the reaching law will fail in the presence of disturbance.
  5. In the control dynamics, the authors never considered the effect of disturbance. The nominal plant assumption is very strange.
  6. Finally, the controller and the observers are designed in different domains. What is the reason behind that? The authors need to focus on either a discrete or continuous form of representation.

Reviewer 3 Report

  • In the introduction, please avoid the use of the term "Literature" to refer to existing papers; it is best to just quote the author's names.
  • What is w (omega) in Equation 6 ?  This needs to be clarified. What is the physical meaning and the units of w ?
  • Please provide a reference for he sliding surface equation shown in equation 10 (exponential reaching law)
  • please explain the meaning and provide a reference of the fhan function (equation 13)
  • what are the units in Equation 15 ? Are f and Fd accelerations, and if yes, what are their physical meanings ?  
  • Similarly, what are the units of x3 in equation 16, and why is x3_dot = w ?  
  • delta 1 and delta 2 in equation 17 were never defined
  • e and delta are not defined in equation 18
  • Please provide a reference to justify the claim of equation (20)
  • what position sensor is used (line 261) and what is its range of travel ?
  • In line 174, it should say "u is the coil current, which is the control input"
  • In Figure 4, what kind of circuit is the power drive ?  Is it a linear amplifier or a PWM amplifier ?  
  • On figure 5, a comparison is presented between the proposed method and "traditional" sliding mode control, but the "traditional" control was never defined in the paper. This needs to be defined.
  • Line 346:  duty cycle was not previously defined. The control action u is defined as current (equation 12), but duty cycle was never defined. Is the control action duty cycle, or current ?
  •  There should be some analysis and comment on what is the expected bandwidth of the proposed controller.

Round 2

Reviewer 1 Report

I have no further comments.

Author Response

Thank you very much for your contribution to this paper.
We have revised and improved the manuscript according to the suggestions.

Reviewer 2 Report

I don't have any further comments. The authors have somehow incorporated my comments in the modified manuscript.  

Author Response

(The authors gave the same response as above.)

Reviewer 3 Report

Response 11 on your letter has not been included on the paper.

Response 13:  It is not possible to calculate analytically  the frequency response of a nonlinear system. But you can numerically use a simulation with input sine waves or noise input to numerically estimate the bandwidth of the closed loop system. This is an essential assessment to illustrate the performance of your controller. Also, bandwidth can be experimentally measured by providing the system with sinusoidal inputs at selected amplitudes.  A discussion of bandwidth with either simulation or measurements is essential in a motion control system.

The main limitation of the paper is that the authors fail to make a case of the need of this paper - what problems are solved by their method that are not previously addressed by other control methods ?  Several references provided in the Introduction claim success in the control of similar systems.  What is the need and the advantage of using a new method ?   This needs to be clearly explained, otherwise there is no need and no clear justification to develop a new method to control linear electromagnetic actuators.

So the main weakness of this paper is that it fails to justify why their solution is better than other control methods used to control the same kind of system.

This paper would be a valuable contribution, if the authors would choose the 3 or 4 best alternative control strategies, and would use either simulations or experiments to demonstrate that their proposed method is better or provides specific advantages compared to the other methods.

IN summary, this is a good paper with potential to be a good contribution to the literature, but it needs to be justified by comparing its performance (perhaps through simulation) to other techniques described in the literature.

An option for the authors is choose the most important methods in the literature, and to clearly state (with performance figures) what are their limitations in performance.  This allows to show (through your experimental results)  that your method has solved a performance limitation that other existing methods have not resolved.
